# Dual-Band Laser Stealth Based on Quasi Photonic Crystals

**Man Yuan** [1], **Jianjing Zhao** [1], **Xinye Liao** [1] and **Xin He** [2,*]

1 Undergraduate School, National University of Defense Technology, Changsha 410073, China; ym18095309945@163.com (M.Y.); 15616175293@163.com (J.Z.); liaoxinye0810@163.com (X.L.)
2 Department of Physics, National University of Defense Technology, Changsha 410073, China
* Correspondence: hexin12a@nudt.edu.cn

**Abstract:** A quasi photonic crystal (QPC) dual-band absorber for laser stealth is designed and numerically studied. It consists of a defective two-dimensional photonic crystal on a thick Ni film. The defective photonic crystal is a continuous Ge layer with air holes, but some of the holes are periodically removed. Under a normal incidence that is perpendicular to the plane of the defects, the absorptivity can achieve 92.8% at the 1.064 µm wavelength and 93.2% at the 1.55 µm wavelength. Within large incident angles (<45 degrees), the dual-band absorptivity is still >80%. Additionally, the emissivity is as low as 5.8%~20.0% and 2.8%~5.8% in the 3–5 µm and 8–14 µm atmospheric windows. It is found that the introduced defects can couple the incidence into the structure and lead to spectral peaks (electromagnetic resonances) even without the bottom Ni film. With the help of the high-loss Ni film, the resonances are transformed into magnetic or/and electric modes of different orders. As a result, the QPC generates four absorption peaks. They are superimposed in pairs, resulting in enhanced absorption of the two laser wavelengths.

**Keywords:** dual-band absorber; photonic crystal; laser stealth





## 1. Introduction

Stealth technology plays a vital role in many military applications. The purpose of stealth is to achieve concealment by aligning the signals of a target with those of the background. If our attention is focused on optical frequencies, we find that it is crucial to realize laser stealth due to the rapid development of laser-guided weapons. In addition, since passive infrared detection systems become more and more powerful at night, reducing the infrared emission of a target will be another key consideration. From these points of view, a target is expected to have high absorption at some laser wavelengths (1.064 µm, 1.55 µm, etc.) and low emission in atmospheric windows (3–5 µm and 8–14 µm).

Traditional materials usually have limitations in terms of multi-functional stealth. For example, multilayer coatings can be used for infrared stealth and achieve low thermal emissivity. However, multilayer coatings tend to result in high reflection of guided lasers and are thus easily locked by laser guidance [1]. To counteract laser guidance, researchers have recently applied frequency selective absorbers based on metal-dielectric-metal structures [2,3] and composite materials, such as Al/ATO [4] and photonic crystals (PCs) [5–22] in stealth technology. Among them, two-dimensional photonic crystals have attracted a great deal of attention due to their manufacturing scalability and high-temperature stability [8].

PCs are artificial optical structures whose refractive indices change periodically with space [9]. Owing to their photonic bandgaps, PCs can be used to control and manipulate the propagation of light. Typically, one-dimensional (1D) PCs are composed of membranes with different refractive indices, and the membranes are arranged periodically in a single direction. Based on 1D PCs, a structure with high absorption (>80%) of the 1.06 µm guided laser has been demonstrated, which also shows color camouflage [5]. An absorber that can reduce the reflection at the wavelength of 10.6 µm (60.2% absorption rate) has been presented [10]. In addition, it has been pointed out that the use of 1D PCs in infrared stealth

in atmospheric windows is feasible [5–7]. Similar to 1D PCs, two-dimensional (2D) PCs have periodic indices in two directions in a plane, while, in another direction perpendicular to the plane, the index is uniform. A usual 2D PC is composed of many parallel dielectric columns. The columns are perpendicular to a plane and are periodically arranged [11].

However, there are still some problems to be solved in laser stealth based on PCs. For example, most absorbers based on PCs have only a single laser absorption band [3,5,10]. They are usually sensitive to the incident angle [12]. The absorption rates at peaks are not high enough [13,14]. Some absorbers are too complex to fabricate [15]. In recent years, researchers have attempted to introduce defects in PCs [9,11,16–21], which provides new ideas for solving the above problems and brings new avenues for the development of stealth technology. Hu et al., studied a truncated PC in the Fano-type interference effect [17] and designed a "hyper-crystal" by introducing hyperbolic metamaterial into 1D PCs [18]. Devi et al., designed a mirror image 1D PC with defects and realized various high-sensitive sensors [19]. Segovia-Chaves et al., introduced linear defects into 2D PCs and studied their influence on electromagnetic propagation [20]. Gamare et al., investigated a PC structure with linear defects and analyzed the effect of defects on photonic bandgaps [21]. Typically, the defects in the literature are in the same plane as the incidence, leading to only resonance [9,11,15,16,19] or waveguide [20,21] effects.

In this work, we designed and numerically investigated a dual-band laser absorber based on a quasi photonic crystal (QPC) structure. The plane of the defects in our absorber is perpendicular to the incidence, which is different from those in the literature [9,11,16–21]. The absorber consists of a 2D Ge/air PC with defects and a bottom metallic film. Due to the introduced defects, the 2D Ge/air PC generates four absorption peaks. With the help of the metallic film, the resonance peaks of the 2D Ge/air PC are redshifted to wavelengths around 1.064 µm and 1.55 µm. Meanwhile, the metallic film can increase optical loss. As a result, the absorption rates at wavelengths of 1.064 µm and 1.55 µm reach 92.8% and 93.2%, respectively. It is worth noting that, the absorber has a very low infrared emissivity in atmospheric windows (5.8%~20.0% at 3–5 µm and 2.8%~5.8% at 8–14 µm). To understand the absorption mechanism, electromagnetic resonances in the QPC are studied. Considering fabrication errors, the influence of structure parameters on absorption features is analyzed. In addition, the effect of the incident angle is briefly discussed.

## 2. Design and Results

Figure 1 shows the schematic view of the designed QPC. Figure 1a illustrates the process of introducing defects in a 2D PC, which obtains the top layer of our QPC. If there are no defects, the 2D PC consists of periodic holes in a Ge layer and the center distance between adjacent holes is 99.4 nm. Figure 1b,c show the structure of the designed QPC. As shown in Figure 1b, it is constructed by a continuous Ni film between a 2D PC with periodic defects and a quartz substrate. The thickness of the Ge layer is $h_1$, and the thickness of the Ni film is $h_2$. Owing to the periodic defects, the QPC has a period of $p$. Each of the holes in the Ge layer has a diameter of $d$. Figure 1c demonstrates that light is incident on the surface of the QPC from free space.

We chose Ge as the top layer because of the following aspects. First, Ge has a relatively high optical loss in the near-infrared, which will be advantageous for the absorption of 1.064 µm and 1.55 µm guided lasers. Second, it has a very high refractive index and can strongly confine the light. This tight confinement can further enhance laser absorption. At last, it is transparent across a broad wavelength range in the mid-infrared, which helps us achieve low emissivity in the atmospheric window. We chose Ni as the bottom metallic layer because it is also high-loss when interacting with the near-infrared light. In our QPC, the Ni film should be thick enough to stop the transmission of the incidence.

For a usual 2D PC composed of periodic cylindrical cavities in metals, if it has no defects, the resonant wavelength is determined by the following equation [22]:

$$\lambda_{ij} = \frac{2\pi(r + \delta(\lambda_{ij})) \cdot n}{\chi_{ij}} \tag{1}$$

where $\lambda_{ij}$ is the resonant wavelength in free space, $r$ is the radius of the hole cavity, $\delta(\lambda_{ij})$ is the skin depth associated with the wavelength, $n$ is the refractive index of the dielectric in the hole cavity, $\chi_{ij}$ is the $j$th root of the derivative of the $i$th Bessel function of the fundamental mode. In the present case, the defects cause the holes to form periodic islands, and each of the islands has 48 holes. This influences the effective refractive index $n$ and the resonant wavelength $\lambda_{ij}$. For ease of design, the 48 holes in each unit cell are 4-fold symmetric.

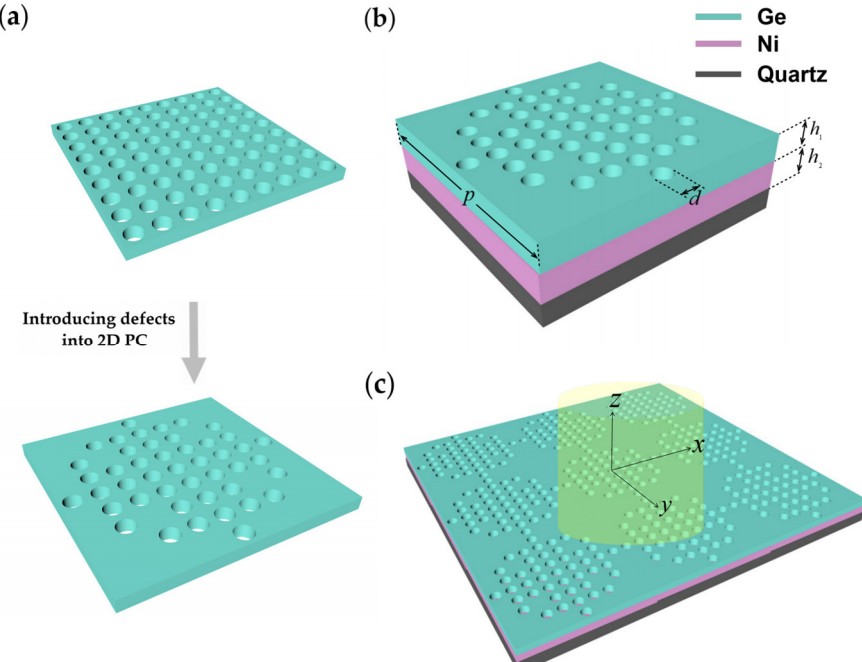

**Figure 1.** Schematic view of the designed QPC. (**a**) Process of introducing defects in the 2D PC; (**b**) The 2D PC with periodic defects seated on a Ni layer and a quartz substrate; (**c**) The structure of a single unit cell of the QPC.

In order to simulate the absorption feature of the designed QPC, a finite difference time-domain (FDTD) method was employed. Periodic boundaries were used in the $x$ and $y$ directions, whereas the boundary conditions in the $z$ direction were set as perfectly matched layers. To improve the accuracy, a fine mesh is used in the computational domain near the holes. In the following simulations, optical constants of Ge and Ni are from the reference [23].

The designed QPC is geometrically symmetric and thus is expected to be polarization independent. Under normal incidence of a transverse-magnetic (TM) polarized light, the reflection, transmission, and absorption spectra of our QPC are plotted in Figure 2. The optimized geometric parameters are $h_1 = 95$ nm, $h_2 = 200$ nm, $d = 96$ nm, and $p = 895$ nm. The absorption (i.e., $A$) is calculated by $A = 1 - R - T$, where $R$ is the reflection and $T$ is the transmission. Figure 2 illustrates that, in the investigated wavelength range, the transmission is almost zero. The reflection at wavelengths of 1.064 μm and 1.55 μm is significantly low. These lead to strong absorption at 1.064 μm and 1.55 μm wavelengths, with absorption rates as high as 93.2% and 95.2%, respectively. The results show that the designed QPC can achieve dual-band guided laser stealth. Additionally, both of the absorption bands around 1.064 μm and 1.55 μm wavelengths have considerable bandwidths. This feature is also advantageous, because guided lasers tend to have wavelength-tuning capability [24].

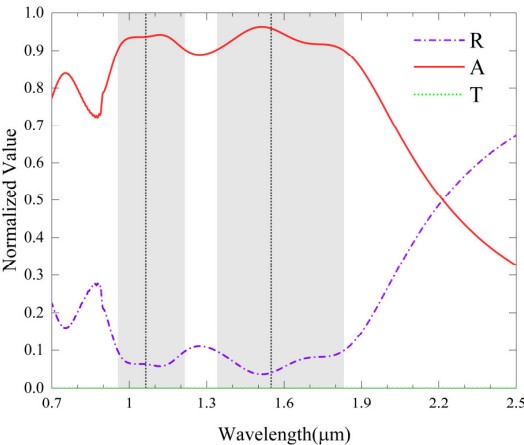

**Figure 2.** Reflection (*R*), transmission (*T*), and absorption (*A*) spectra of the QPC under normal incidence. The wavelengths of 1.064 μm and 1.55 μm are marked by two vertical dashed lines, respectively. The two shaded ranges indicate the absorption of >90% near the 1.55 μm and 1.064 μm wavelengths.

## 3. Absorption Mechanism

To understand the mechanism of high absorption at wavelengths of 1.064 μm and 1.55 μm, we first investigated the effect of the defects on the absorption feature. Figure 3 shows the absorption spectra corresponding to a usual 2D PC, a usual 2D PC with bottom Ni film, a defective 2D PC, as well as our QPC, respectively. Here, the structural evolution keeps the material and the other geometric parameters the same as those in our QPC. It can be seen that, for a usual 2D PC, the absorption in the interested wavelength range is low and there are almost no absorption peaks. Adding a Ni film to the bottom of a usual 2D PC only increases broad-spectrum absorption but generates no absorption peaks. On the contrary, introducing defects to a usual 2D PC produces obvious absorption peaks ($\lambda_{1a}$, $\lambda_{2a}$, $\lambda_{3a}$, $\lambda_{4a}$). A comparison between the absorption spectra of a usual 2D PC and a defective 2D PC clearly indicates that the peaks are attributed to the defects. For these reasons, our QPC has four obvious absorption peaks ($\lambda_{1b}$, $\lambda_{2b}$, $\lambda_{3b}$, $\lambda_{4b}$), and they result from the defects in the top layer. According to Equation (1), the resonance wavelength will increase with increased refractive index. The Ni film in our QPC enlarges the effective index, thus causing the original peaks to redshift. Moreover, the absorption rates and bandwidths all exhibit significant increases due to the high loss of the Ni film. Therefore, our QPC can strongly absorb the 1.064 μm and 1.55 μm guided lasers.

The results also imply that the four absorption peaks ($\lambda_{1b}$, $\lambda_{2b}$, $\lambda_{3b}$, $\lambda_{4b}$) of our QPC seem to be related to the peaks ($\lambda_{1a}$, $\lambda_{2a}$, $\lambda_{3a}$, $\lambda_{4a}$) of the defective 2D PC. To verify this, we studied the electromagnetic resonances in our QPC and in the defective 2D PC. Since our structure is symmetric, TM polarized incidence was used in the computations. In the following, we will first investigate the two resonances near the 1.55 μm wavelength, and then the other two resonances near the 1.064 μm wavelength. The relations between the resonances in our QPC and those in the defective 2D PC will be explored.

Figure 4 illustrates the resonance characteristics related to the two absorption peaks near the 1.55 μm wavelength. For the defective 2D PC, the displacement current and magnetic field distributions at $\lambda_{1a}$ are shown in Figure 4a,b, respectively. It is observed that the displacement current is in the *x* direction and the $H_z$ in the upper defects is opposite to that in the lower defects. Thus, a circular magnetic field exists, and resonance $\lambda_{1a}$ is mainly attributed to electric resonance in the defects. Regarding our QPC at $\lambda_{1b}$, the displacement current in the Ni film is opposite to that in the Ge layer, as shown in Figure 4c. Owing to the Ni film, three current circuits are formed. Combining these current circuits with the $H_y$ in Figure 4d, it is clear that each current circuit leads to a magnetic moment in the *y* direction. These suggest that magnetic resonance occurs at $\lambda_{1b}$. For clarity, Figure 4e plots a schematic diagram of the magnetic resonance. Resonance $\lambda_{1b}$ corresponds to a fundamental magnetic mode and is mainly confined to the defects.

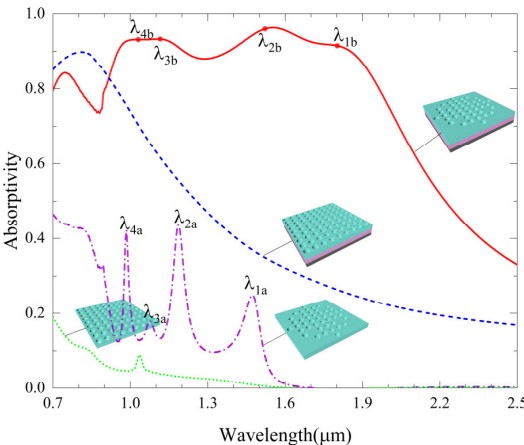

**Figure 3.** The effect of structural evolution on the absorption spectrum, where the cases of a usual 2D PC, a usual 2D PC with bottom Ni film, a defective 2D PC, and our QPC are shown. The absorption peaks associated with the defective 2D PC are marked by $\lambda_{1a} = 1.47$ μm, $\lambda_{2a} = 1.185$ μm, $\lambda_{3a} = 1.079$ μm, and $\lambda_{4a} = 0.985$ μm, respectively. The absorption peaks associated with our QPC are marked by $\lambda_{1b} = 1.802$ μm, $\lambda_{2b} = 1.527$ μm, $\lambda_{3b} = 1.119$ μm, and $\lambda_{4b} = 1.029$ μm, respectively.

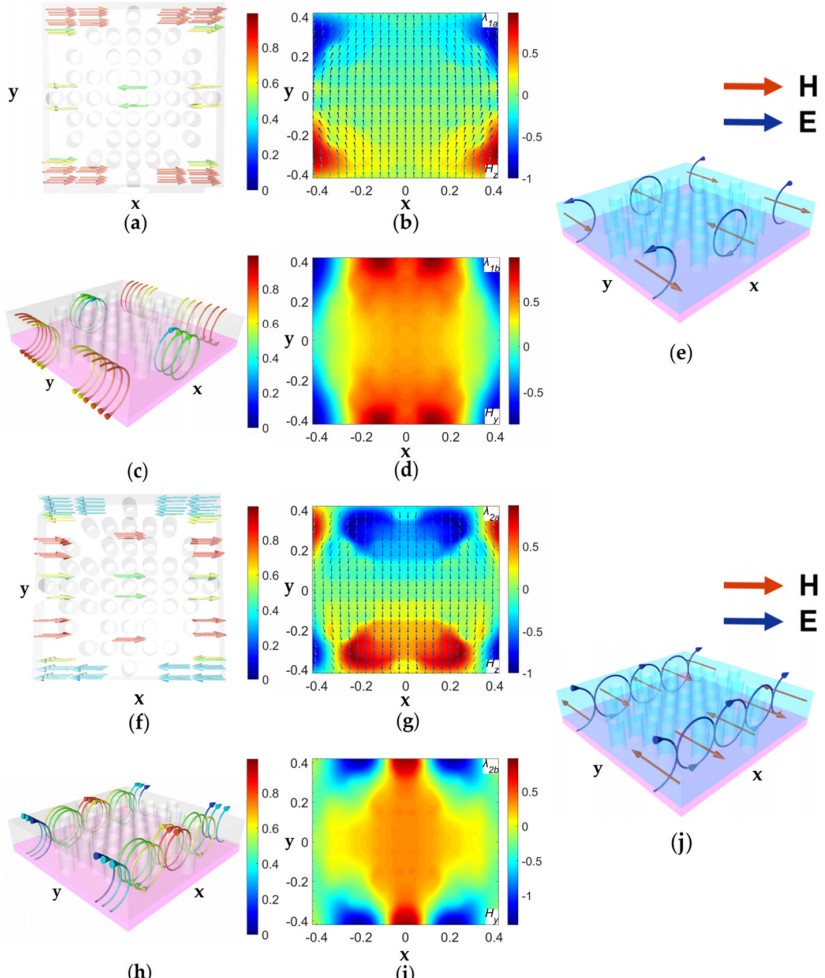

**Figure 4.** Resonance characteristics related to the two peaks near the 1.55 μm wavelength. (**a**) Displacement current and (**b**) Magnetic field distributions at $\lambda_{1a}$; (**c**) Displacement current, (**d**) Magnetic field, and (**e**) Schematic resonance at $\lambda_{1b}$; (**f**,**g**) are the same as (**a**,**b**) but are at $\lambda_{2a}$; (**h**–**j**) are the same as (**c**–**e**) but are at $\lambda_{2b}$. Note that the arrows in (**b**,**g**) represent *H* vectors. The color maps in (**b**,**d**,**g**,**i**) correspond to a certain dominant component as marked, respectively.

Using a similar method to analyze resonance $\lambda_{2a}$, we can find that there are two circular magnetic fields in each unit cell of the defective 2D PC and resonance $\lambda_{2a}$ is also attributed to electric resonance, as shown in Figure 4f,g. As for our QPC structure at $\lambda_{2b}$, three current circuits are observed, as shown in Figure 4h. Combining these current circuits with the $H_y$ in Figure 4i, it is indicated that magnetic resonances occur. The schematic diagram of the magnetic resonances is illustrated in Figure 4j. It is known that resonance $\lambda_{2b}$ corresponds to a second-order magnetic mode.

Figure 5 shows the resonance characteristics associated with the two absorption peaks near the wavelength of 1.064 µm. For the defective 2D PC, Figure 5a,b suggests that the resonance $\lambda_{3a}$ in the defective 2D PC is mainly attributed to electric resonance in the defects. Because $\lambda_{3a}$ is relatively shorter, this resonance is of higher order. For our QPC at $\lambda_{3b}$, Figure 5c,d demonstrates that a second-order magnetic resonance occurs due to the bottom Ni film. It is worth noting that the $H_z$ in the upper defects is opposite to that in the lower defects, as shown in Figure 5e. After a close examination of Figure 5c,e, it is found that there are also electric resonances. Figure 5f plots the electromagnetic resonances schematically. A second-order magnetic resonance is superimposed on a fundamental electric resonance, and they result in the absorption peak at $\lambda_{3b}$.

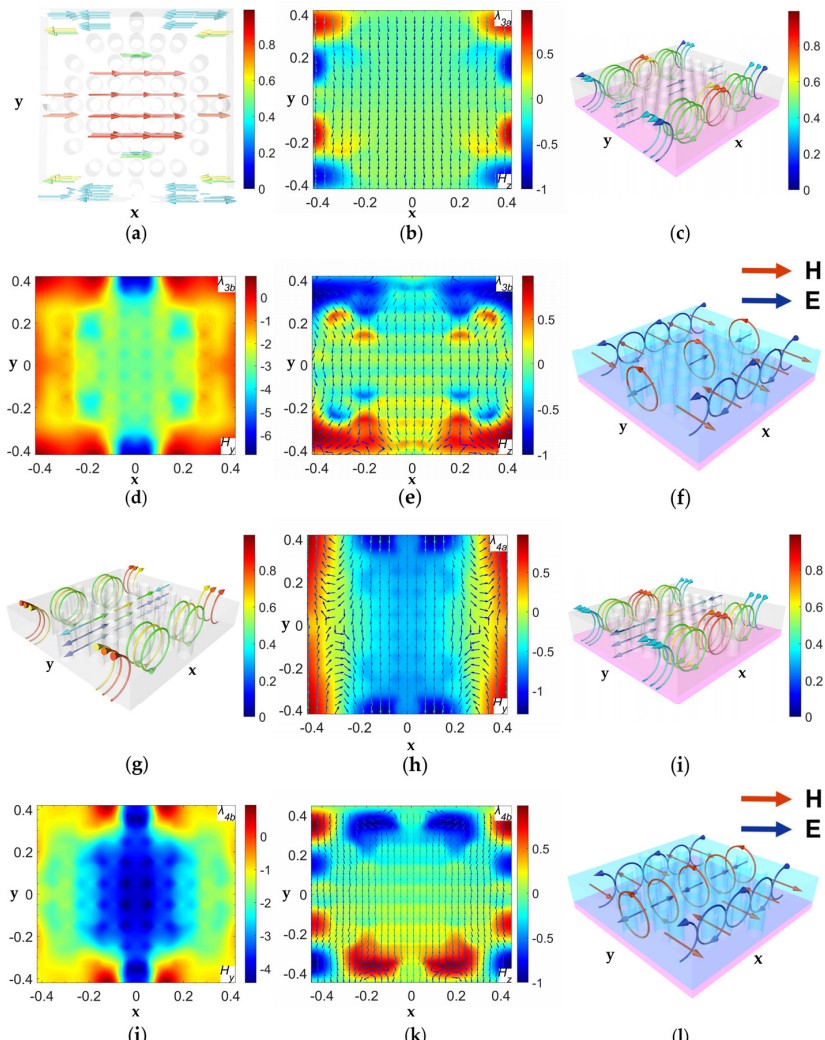

**Figure 5.** Resonance characteristics related to the peaks near the 1.064 µm wavelength. (**a**) Displacement current and (**b**) Magnetic field distributions at $\lambda_{3a}$; (**c**) displacement current; (**d**,**e**) magnetic field, and (**f**) schematic resonance at $\lambda_{3b}$; (**g**,**h**) are the same as (**a**,**b**) but are at $\lambda_{4a}$; (**i**–**l**) are the same as (**c**–**f**) but are at $\lambda_{4b}$; the arrows in (**b**,**e**,**h**,**k**) represent the $H$ vectors. The color maps in (**b**,**d**,**e**,**h**,**j**,**k**) correspond to a certain dominant component as marked, respectively.

However, as for the resonance $\lambda_{4a}$ in the defective 2D PC, the phenomenon is different. According to Figure 5g,h, the directions of the displace currents in the defective area are opposite on the top and bottom Ge surfaces, whereas the direction of $H_y$ is unchanged. These suggest that the top and bottom surfaces of the Ge layer reflect the light to form Fabry-Pérot resonance. For resonance $\lambda_{4b}$, the displacement currents and the $H_y$ in the QPC become complicated because of the Ni film, as shown in Figure 5i,j. We can observe four magnetic resonances in each unit cell. A comparison between Figure 5j,k indicates that higher-order electric resonances also exist. Figure 5l schematically illustrates the electromagnetic resonance at $\lambda_{4b}$. Obviously, a second-order electric resonance and a second-order magnetic resonance together lead to the peak at $\lambda_{4b}$.

The results in Figures 4 and 5 can be concluded as follows. For the defective 2D PC, the peaks ($\lambda_{1a}$, $\lambda_{2a}$, $\lambda_{3a}$, $\lambda_{4a}$) mainly result from electric resonances of different orders. However, because of the Ni film in our QPC, these electric resonances are transformed into magnetic (and electric) resonances of different orders, leading to the four absorption peaks at $\lambda_{1b}$, $\lambda_{2b}$, $\lambda_{3b}$, and $\lambda_{4b}$. The transformation occurs because, compared with the defective 2D PC, there exists an induced displacement current in the Ni film and its direction is opposite to that in the top 2D PC layer.

## 4. Discussion

In practical applications, the performance of the designed QPC will be affected by structural errors. To account for this, absorption spectra were simulated by changing the geometric parameters $p$, $d$, and $h_1$. In the calculations below, only a single parameter is changed independently, while the other parameters remain the same as those used in Figure 2. For clarity, the wavelengths of 1.064 μm and 1.55 μm are marked by vertical dashed lines, respectively.

Based on the absorption mechanism mentioned above, it is known that changing the period will affect the electromagnetic resonance in the defects and hole islands. Figure 6 shows the calculated absorption spectra when adjusting the period of the QPC. It demonstrates that the four resonance peaks all redshift with the increase of the period. Meanwhile, the absorption rate of each peak degrades to a different extent if the period enlarges. When the period alters from 895 nm to 1050 nm, the absorption rate at the wavelength of 1.064 μm reduces from 93.7% to 78.5%, and the absorption rate at the wavelength of 1.55 μm decreases from 96.5% to 84.4%. However, if the change in period is less than 20% relative to the optimized parameter, the absorption at 1.064 μm and 1.55 μm can still exceed 80%.

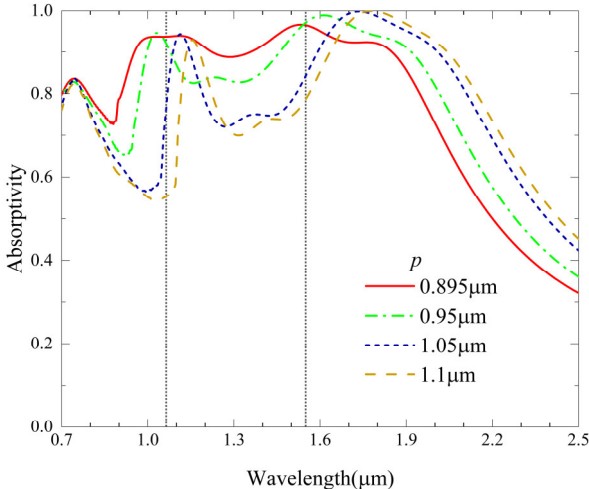

**Figure 6.** Absorption spectra with different periods. The solid line corresponds to the absorption cure in Figure 2.

Figure 7 plots the calculated absorption curves when changing $h_1$. It should be noted that different $h_1$ means different hole depths. It is shown that the four resonance peaks blue

shift with the decrease in $h_1$. This is consistent with Equation (1), because the refractive index becomes smaller with reduced $h_1$. As $h_1$ decreases, the absorption rate of each peak exhibits degradation due to the weak coupling of incident light into the top layer. When $h_1$ increases from 95 nm to 115 nm, the absorption rates at 1.064 μm and 1.55 μm wavelengths only slightly decrease. However, if $h_1$ decreases from 95 nm to 75 nm, there is a larger decrease in the absorption rate at 1.064 μm wavelength (from 96.5% to 88.3%).

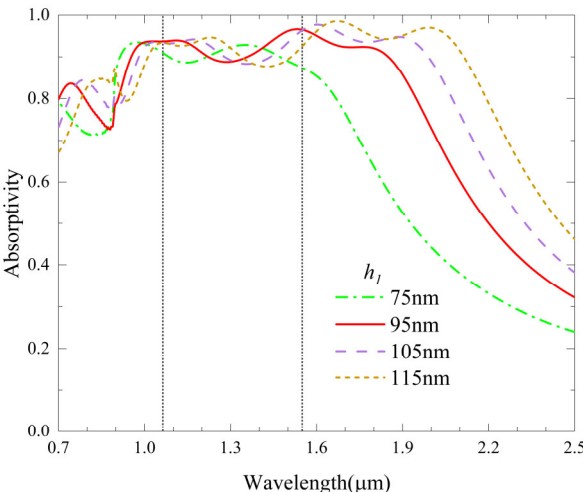

**Figure 7.** Absorption spectra with different thicknesses of the Ge layer.

The absorption feature associated with different hole diameters is illustrated in Figure 8. We can observe that the four resonance peaks blue shift with the increase in $d$. The reason is as follows. According to Equation (1), the $n$ in our QPC depends not only on the air holes but also on the defects. The larger the $d$, the larger the $r$ in Equation (1), but it compresses the defects, thus reducing $n$ in Equation (1). As a result, the overall effect is to reduce the resonant wavelength. Regarding the change in the peak absorption, similar phenomena are seen, which also result from the coupling strength of the incidence. The results indicate that the performance of our QPC is relatively sensitive to the hole diameter. If $d$ goes down by 10 nm relative to the optimal parameter, the absorption rate at the wavelength of 1.064 μm reduces from 93.7% to 80.8%. However, a smaller $d$ can improve the absorption rate at the wavelength of 1.55 μm (99.5%).

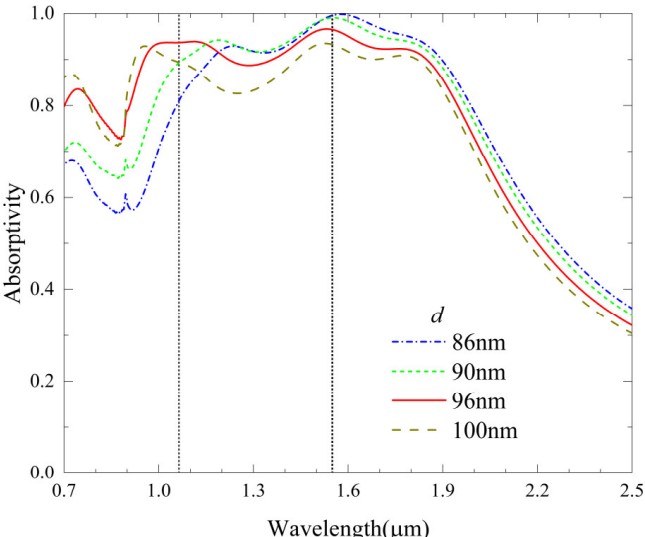

**Figure 8.** Absorption spectra as a function of the hole diameter $d$.

In practical conditions, we may not know the angle from which the guided laser is coming. Some researchers pointed out that only normal incidence and reflection need to be considered [25]. However, we believe that it would be very advantageous to have high absorption rates in a large range of incident angles. Therefore, we simulated the performance of the designed QPC under oblique incidence, and the results are shown in Figure 9. Since the QPC is polarization insensitive, we have assumed the incident laser is TM polarized. As can be seen, when the laser is tilted in the plane of incidence and polarization (Figure 9a), the absorption rate at the wavelength of 1.064/1.55 μm decreases/increases. If the laser is tilted in the plane of incidence (Figure 9b), both the absorption rates at 1.064 μm and 1.55 μm wavelengths decrease. As long as the incident angle is within ±45 degrees, the absorption rates at 1.064 μm and 1.55 μm wavelengths are still higher than 80%. Therefore, the QPC has the advantage of wide-angle absorption of guided lasers.

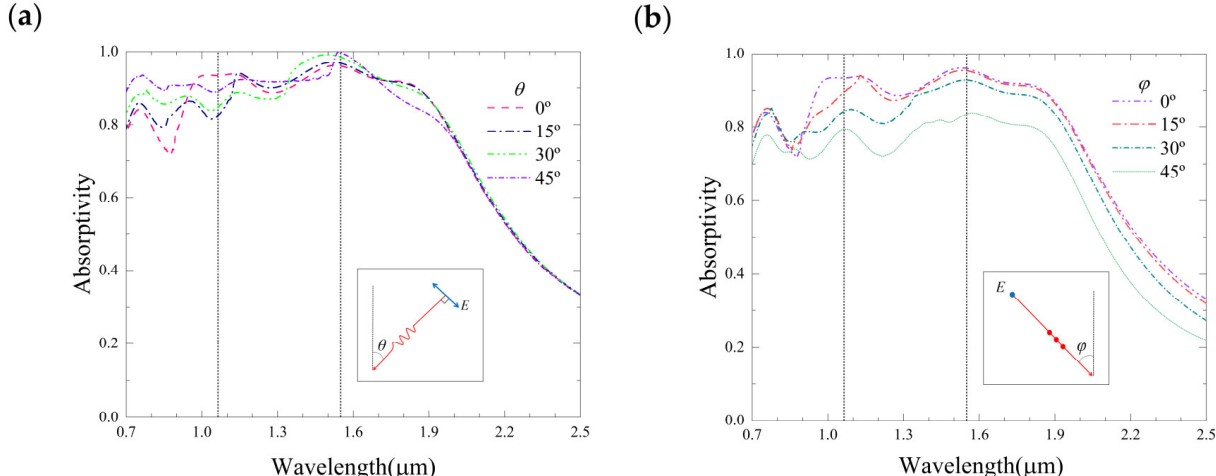

**Figure 9.** Absorption spectra under oblique incidence. (**a**) Polarization is in the incident plane; (**b**) Polarization is perpendicular to the incident plane.

As mentioned above, it is highly desired that a laser stealth surface can achieve low emission in atmospheric windows. To simply show the infrared stealth performance of the QPC, we assume that a blackbody target with/without the QPC has different temperatures (283 K and 313 K) during the day and night, respectively. According to Kirchhoff's law, the emission of the QPC $\sigma(\lambda, T)$ can be calculated by

$$\sigma(\lambda, T) = A(\lambda) \cdot u(\lambda, T) \tag{2}$$

where $A(\lambda)$ is the absorption rate of the QPC, $u(\lambda, T)$ is the Planck blackbody function. Figure 10a plots the simulated absorption feature of the QPC at mid-infrared frequencies. It is seen that the QPC has a low absorption rate of <20.0% in the 3–5 μm band and <5.8% in the 8–14 μm band, respectively. Figure 10b demonstrates the emission power of the blackbody and the QPC at 283 K (night) and 313 K (day). Obviously, the emission of the QPC in the mid-infrared is significantly lower than that of the blackbody at the same temperature. It must be noted that there is sunlight during the day. The QPC can reflect the sunlight, which will considerably degrade its performance of infrared stealth during the day. Therefore, the QPC has a good capability of avoiding passive infrared detection at night.

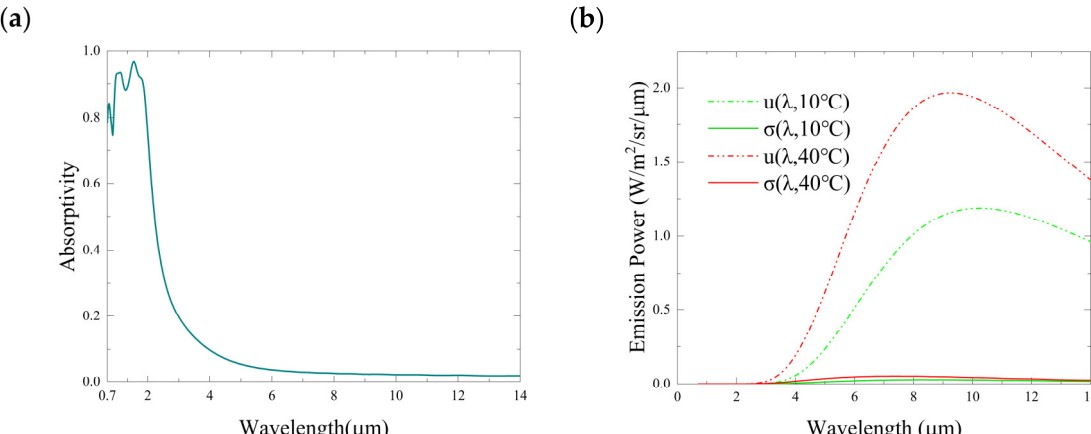

**Figure 10.** Simulation results of infrared stealth performance. (**a**) Calculated broadband absorption of the QPC; (**b**) Blackbody emission and QPC emission at different temperatures.

## 5. Conclusions

We have designed a QPC structure consisting of a defective 2D PC and a Ni film. The defective 2D PC is constructed by a Ge layer with periodic arrays of air holes, some of which are missing. Numerical results show that the QPC can strongly absorb guided lasers at wavelengths of 1.55 μm and 1.064 μm, and can be used in laser stealth applications. At normal incidence, the absorption rates can achieve 93.2% and 92.8%, respectively. Based on FDTD simulations, it is found that absorption peaks can be generated even without the Ni film and they are due to the introduction of the defects. After a detailed study of the resonances in the defective PC and our QPC, it is revealed that the existence of the Ni film in our QPC transforms the electric resonances in the defective 2D PC into electromagnetic resonances of different orders. These transformed resonances, as well as the high loss of Ni, lead to four absorption peaks at 1.802 μm, 1.527 μm, 1.119 μm, and 1.029 μm. The two resonances at wavelengths of 1.802 μm and 1.527 μm are superimposed to form high absorption of the 1.55 μm guided laser, while the two resonances at wavelengths of 1.119 μm and 1.029 μm are superimposed to form high absorption of the 1.064 μm guided laser. When changing the geometric parameters, the absorption performance of the QPC can be maintained well. At a large incident angle of 45 degrees, the dual-band absorption can still be >80%, which will be advantageous for practical applications. In addition, the QPC has considerably low absorption rates in the 3–5 μm and 8–14 μm bands, meaning that the QPC also has the capability of infrared stealth. This work offers a kind of PC-based absorbers that is useful in military camouflage, laser technology, and other fields.

**Author Contributions:** Conceptualization, M.Y. and X.H.; methodology, M.Y.; software, M.Y., J.Z. and X.L.; validation, X.H.; formal analysis, X.H.; investigation, M.Y. and X.L.; resources, M.Y.; data curation, M.Y.; writing—original draft preparation, M.Y.; writing—review and editing, X.H.; visualization, J.Z. and M.Y. All authors have read and agreed to the published version of the manuscript.

**Funding:** This research was funded by the National Natural Science Foundation of China (Nos. 12272407, 62275271, and 62275269), the National Key Research and Development Program of China (2022YFF0706005), the Natural Science Foundation of Hunan Province (No. 2022JJ40552, and 2020JJ5646), and the Program for New Century Excellent Talents in University (NCET-12-0142).

**Institutional Review Board Statement:** Not applicable.

**Informed Consent Statement:** Not applicable.

**Data Availability Statement:** The data presented in this study are available on request from the corresponding author.

**Conflicts of Interest:** The authors declare no conflict of interest.

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
