# Peer review of "Dual-Band Laser Stealth Based on Quasi Photonic Crystals"

_photonics, doi:10.3390/photonics10080931_

Round 1

Reviewer 1 Report

In this manuscript, the authors investigated the quasi photonic crystal dual-band absorber for laser stealth. The manuscript is well prepared. The method is scientifically correct, and the results and discussion can support the conclusion. In my option, the manuscript can be published in this journal.

I have several comments on the manuscript.

1. Compared to other research works on stealth technology using defects in PCs (Refs. 9,11,15-19), what is the novelty of this work.

2. It would be better if the authors could add more description about the defect of the structure.

2. Some pictures are not clear enough.

3. Which software did the authors use. Is there any license that need to be provided.

4. There are some typos in the manuscript.

Minor editing of English language required.

Reviewer 2 Report

In this manuscript, the authors designed a kind of QPC (quasi photonic crystal) dual-band absorber composed of a defective 2D PC and a Ni film. The QPC could exhibit high absorptivity within two bands while remaining low emissivity in atmospheric windows, which may be useful in military applications, especially laser stealth. The mechanism of absorption peaks was discussed by analyzing electromagnetic field results. Moreover, the parametric influence was minutely studied, which further proved the performance of proposed QPC structure. I do recommend the acceptance of this paper after the authors properly revise the manuscript based on the following points:

1. Could the authors give more details about how the transformation from electric resonances to magnetic ones results in absorption peaks.

2. The visualization of figures should be further optimized. For example,

a) In spectrum figures, two shaded ranges may help to indicate the “dual-band” property;

b) In Figs. 4 and 5, the arrows, the font sizes (for axis, ticks and labels) and the observed perspectives should be optimized.

3. The description and analysis for Figs. 4 and 5 seem a little wordy, which may be organized in a more concise manner.

4. Recently, the omnidirectional absorbers are realized with hyperbolic metamaterials, these works should draw the authors’ attention [Enhanced Magneto-Optical Effect in Heterostructures Composed of Epsilon-Near-Zero Materials and Truncated Photonic Crystals, Frontiers in Materials. 9, 843265 (2022); Photonic Dirac Nodal Line Semimetals Realized by Hyper-crystal. Physical Review Research 4(2), 023047 (2022)].

Minor editing of English language required

Reviewer 3 Report

Reviewer comments:

1. In this Paper its mention that metalic film has helped the resonance peak of the 2D Ge/air PC are around 1.064 and 1.55 um. How it will happen? need some points. 2. How metalic film cause increase in metalic loss? 3. Quartz substrate is only used to support the structure and it can be replaced by other materials. Is there any other substrate to support? If Yes how quartz substrate is different? 4. Need small explantion on how wavelength range will affect the absorption peak. 5. Author has mention that direction of Displacement current varies H & Hz. Need some clarity. And also how upper defects in opposite direction. 6. Absorption rate at the wavelength of 1.064 μm reduces from 93.7% to 78.5%. whether it is possible yo reduce more? 7. The designed QPC has a good capability of avoiding passive infrared detection at night. Need some clarity on this and data required day & night variation. 8. Whether incident angle of 45 degree is enough for practical application? Which angle is perfect for that and how it can be achieved?

Minor revisions 

Round 2

Reviewer 3 Report

All the necessary changes have been made and the manuscript can be accepted

Minor spell check required